# Tick Infestation and Molecular Detection of Tick-Borne Pathogens from Indian Long-Eared Hedgehogs (*Hemiechinus collaris*) in Pakistan

**DOI:** 10.3390/ani14223185

**Published:** 2024-11-06

**Authors:** Shahzad Ali, Michael E. von Fricken, Asima Azam, Ahmad Hassan, Nora G. Cleary, Kiran Iftikhar, Muhammad Imran Rashid, Abdul Razzaq

**Affiliations:** 1Wildlife Epidemiology and Molecular Microbiology Laboratory (One Health Research Group), Discipline of Zoology, Department of Wildlife and Ecology, University of Veterinary and Animal Sciences, Ravi Campus, Lahore 54000, Pakistan; ahmaduvas1@gmail.com; 2Department of Environmental and Global Health, University of Florida, Gainesville, FL 32610, USA; ncleary@phhp.ufl.edu; 3Department of Zoology, Shaheed Benazir Bhutto Women University, Peshawar 25000, Pakistan; asimaazam@sbbwu.edu.pk; 4Department of Mathematics and Statistics, University of Agriculture Faisalabad, Faisalabad 38000, Pakistan; kiran.iftikhar@uaf.edu.pk; 5Department of Parasitology, University of Veterinary and Animal Sciences, Lahore 54200, Pakistan; imran.rashid@uvas.edu.pk; 6Animal Sciences Division, Pakistan Agricultural Research Council, Islamabad 44000, Pakistan; abdulrazzaqazrc@gmail.com

**Keywords:** *Anaplasma* spp., *Babesia* spp., *Theileria* spp., Pakistan

## Abstract

Hedgehogs play a significant role as a vertebrate host for ticks and the tick-borne pathogens (TBPs) they carry, which include bacterial, viral, and parasitic diseases of concern for both humans and livestock. Not only are TBPs a major health concern, but they also directly impact the economy and livelihood of farmers and pastoral systems that rely on livestock. This study was conducted to molecularly confirm tick species infesting hedgehogs and to detect TBPs in both ticks and hedgehogs. The DNA of ticks and the blood of the hedgehogs were extracted and analyzed. Based on the sequence results, *Rhiphicephalus turanicus* was identified as the only tick species in this study. *Anaplasma marginale*, *Babesia bigemina*, and *Theileria lestoquardi* were detected in the ticks, and *A. marginale* was detected in the blood of one hedgehog. Phylogenetic analysis revealed that these pathogens were highly similar to sequences from cattle blood reported in Thailand, the United States, India, South Africa, and Uganda. By understanding genetic differences and similarities, improved measures to monitor for TBPs as well as interventions that aim to protect human and animal populations can be implemented.

## 1. Introduction

Ticks are well-known vectors of disease and have been recorded on numerous domestic and wild animals, [1,2] transmitting pathogens including bacteria (*Anaplasma* spp., *Borrelia* spp., *Coxiella* spp., *Ehrlichia* spp., and *Francisella* spp.), viruses (Crimean-Congo Haemorrhagic Fever virus, and Tick-Borne Encephalitis virus), and parasites (*Babesia* spp. and *Theileria* spp.) [3,4]. In Pakistan, tick genera, *Haemaphysalis*, *Hyalomma*, *Ornithodoros*, and *Rhipicephalus*, have been reported to infest humans and animals [5], with *Hyalomma*, *Rhipicephalus*, and *Haemaphysalis* genera most commonly found in Pakistan’s Punjab province [6].

There are three major tick-borne diseases (TBDs), anaplasmosis, babesiosis, and theileriosis, which pose a significant risk to livestock in Pakistan [7]. Anaplasmosis is distributed worldwide in tropical and subtropical areas often in bovines and is mainly caused by *Anaplasma marginale* followed by *A. centrale.* Babesiosis, a disease of veterinary and human importance, can be caused by *Babesia bovis* and *B. bigemina* in livestock. Theileriosis, which is found throughout Africa and Asia, is responsible for severe economic losses in cattle, water buffaloes, sheep, goats, and horses [8,9]. The main vectors of these diseases in Pakistan are *Rhipicephalus* spp. and *Hyalomma* spp. All three diseases can result in reduced animal production and decreased meat and milk yield, ultimately affecting the livelihood of farmers with a mortality rate upwards of 90% from babesiosis and theileriosis. The surveillance of tick-infested animals in Pakistan is vital to monitor the threat of TBDs for veterinary health [6].

The Indian long-eared hedgehog (*Hemiechinus collaris*), a member of the Chordata and Mammalia phyla, the Eulipotyphla order, and the Erinaceidae family, is native to Central Asia and parts of the Middle East. Hedgehogs play an important role in the transmission and maintenance of zoonotic pathogens as they serve as reservoirs and ectoparasite hosts to a range of diseases and arthropod vectors [10]. Tick species from the genera *Hyalomma*, *Amblyomma*, *Haemaphysalis*, *Dermacentor*, *Ornithodoros*, and *Rhipicephalus* have been recorded on hedgehogs [11,12]. Tick-infested hedgehogs are known to harbor *Borrelia* spp., *Rickettsia felis*, *A. marginale*, and *A. phagocytophilum* [12,13,14]. The domestication of hedgehogs has become more frequent in recent decades, resulting in greater animal–human contact, therefore increasing the threat of TBP spillover to people from hedgehogs. While hedgehogs are widely distributed in Pakistan, especially in the South, there is limited information regarding ticks and TBPs of hedgehogs. The characterization of these pathogens and their vectors is critical to improving our understanding of their reservoir hosts, distribution, and the potential threat they pose to human and animal health. The present study aims to determine the prevalence and molecular confirmation of tick species and TBPs associated with Indian long-eared hedgehogs in Pakistan.

## 2. Materials and Methods

### 2.1. Study Site

This study was conducted in the Sahiwal division (which contains three districts: Sahiwal, Okara, and Pakpattan) of Punjab province in Pakistan from December 2019 to December 2020 (Figure 1). Sahiwal has a total area of 10,302 km^2^. This region has extreme weather conditions due to substantial temperature variation, a low of 5 °C during winter and a high of 50 °C in the summer, along with an annual average rainfall of 349 mm [15]. Sahiwal division is recognized for its livestock production, with a focus on rearing Sahiwal cattle breeds and Nili-Ravi buffaloes. In addition to these breeds, sheep and goats are also raised in this area. Unfortunately, these livestock species are susceptible to various ticks and TBPs [16,17].

### 2.2. Sampling and Data Collection

Live cage traps and hand nets were used to collect Indian long-eared hedgehogs (*Hemiechinus collaris*). Hedgehog traps were set at dusk, baited with peanut butter and apples, and examined for captured hedgehogs at dawn. The entire body of the hedgehog was examined for ticks, which were removed using a delicate tweezer without damaging tick mouth parts. The ticks were then preserved in sealed bottles containing 70% ethanol following collection, as previously described [18]. Blood samples were collected from the jugular vein of each tick-infested hedgehog using a 3 mL syringe fitted with a 22-gauge needle [19]. The field data, including sex, age, urbanicity, habitat, and district, were recorded. The hedgehogs were released back to their natural habitat after the collection of samples and data.

### 2.3. Tick Identification

A total of 109 ticks were morphologically identified using taxonomic keys under a stereomicroscope (Euromex, Duiven, The Netherlands) as to the sex, developmental stage (nymph/adult), and species level [20].

### 2.4. DNA Extraction from Ticks and Blood

The ticks were homogenized in liquid nitrogen using a mortar and pestle and then placed into a 1.5 mL microcentrifuge tube. DNA was extracted from the homogenized ticks (20 mg) and hedgehog blood (200 µL) using the Thermo Scientific GeneJET genomic DNA purification kit (Thermo Fisher Scientific, Vilnius, Lithuania) according to manufacturer instructions. DNA concentration was assessed with the Nanodrop 2000 spectrophotometer (Thermo Fisher Scientific, Waltham, MA, USA). Purified DNA was kept at −20 °C till further use.

### 2.5. PCR Confirmation of Tick Species

Tick DNA was detected by the amplification of the intergenic spacer 2 (ITS2) region (710 bp) using the following primers: ITS-2F (5′-AGGACACACTGAGCACTGATTC-‘3) and ITS-2R (5′-ACTGCGAAGCACTTRGACCG-‘3) for the molecular identification of tick species [21]. The PCR mixture was made of 10 µL master mix (WizBio, Seongnam-si, Republic of Korea), 2.5 µL of template DNA, 1 µL of forward primer, 1 µL of reverse primer (Macrogen, Seoul, Republic of Korea), and 10.5 µL of DEPC water (Thermo Fisher Scientific, Waltham, MA, USA) for a total volume of 25 µL. The PCR was run for 35 cycles with the first denaturation for 5 min at 95 °C, the second for 30 s at 95 °C, annealing for 30 s at 57 °C, extension for 30 s at 72 °C, and the final extension for 5 min at 72 °C [21].

### 2.6. PCR Screening for Microbes

The genus-specific primers corresponding to msp1b, 18S rRNA, and cytb regions were used for the detection of *Anaplasma* spp., *Babesia* spp., and *Theileria* spp. in ticks and blood, respectively (Table 1). The PCR master mix consisted of 12 µL master mix (WizBio, Seongnam-si, Republic of Korea), 5 µL of template DNA, 1 µL of forward primer, 1 µL of reverse primer (Macrogen, Seoul, Republic of Korea), and 6 µL of DEPC water (Thermo Fisher Scientific, Waltham, MA, USA) for a total volume of 25 µL. The amplification of target regions was confirmed on a 1.5% Agarose gel (Bio World, Visalia, CA, USA) using electrophoresis. The PCR results were visualized with Bio-Rad gel visualization equipment (Bio-Rad, Berkeley, CA, USA).

### 2.7. Sequencing and Statistical Analysis

The PCR-positive samples were sequenced by Microgen, Seoul, Korea. All sequences were trimmed with a biological sequence alignment editor (BioEdit 7.2). To compare the identity of each tick and TBPs, reference sequences were obtained from GenBank. The sequences were aligned in Geneious Prime 2022.11.0.14.1 using the MAFFT algorithm. Maximum likelihood phylogenetic trees were created in MEGA 10.2.6 using the best match substitution model for each corresponding pathogen with bootstrap repetitions set at 1000 [24,25]. The bootstrap values under 50% were excluded from the phylogenetic trees. Statistical analysis was performed using the SPSS version 21 statistical tool (IBM Armonk, New York, NY, USA). A host with a single tick present was classified as infested, while a host with no ticks detected was considered non-infested. The statistical significance of tick and tick-borne pathogens prevalence was determined using a Chi-square test with significance set at 5% [26].

## 3. Results

### 3.1. Tick Infestation

From December 2019 to December 2020, 64 hedgehogs were captured from the Sahiwal division and 16 (25%) were infested with ticks. Female hedgehogs had a higher tick infestation (29.6%) compared to male hedgehogs (21.6%). A higher number of adult hedgehogs were captured than juvenile hedgehogs, but juvenile hedgehogs had a higher prevalence of ticks (31.2%) than adults (22.9%). A similar number of hedgehogs captured in rural areas (26.3%) and urban areas (23.1%) were found to be infested with ticks. The prevalence of ticks was also similar on hedgehogs found in agricultural land (25.7%) and animal farms (24.1%). Tick prevalence was comparable in all three districts: Okara (26.1%), Pakpattan (25%), and Sahiwal (23.8%). No significant difference was found among these factors for the prevalence of tick infestation (*p* > 0.05) (Table 2). A total of 109 *R. turanicus* ticks were collected from long-eared hedgehogs. Male ticks were more prevalent (67.9%; *n* = 74) compared to female ticks (32.1%; *n* = 35). The ticks were in two life stages: adult (91.7%; *n* = 100) and nymph (8.3%; *n* = 9) (Table 3).

### 3.2. Prevalence of TBPs in Hedgehog Blood and Ticks

*Anaplasma marginale* was detected in an adult male hedgehog captured from a rural area of district Sahiwal (Table 4). Out of 109 ticks, 10 (9.2%) were positive for TBPs such as *T. lestoquardi* (3.7%), *A. marginale* (2.8%), and *B. bigemina* (2.8%). A higher number of male ticks were infected with *T. lestoquardi* (2.7%) and *A. marginale* (2.7%) compared to *B. bigemina* (1.4%). However, female ticks had higher infection rates of *T. lestoquardi* (5.7%) and *B. bigemina* (5.7%) than *A. marginale* (2.9%). Adult ticks were comparatively more infected with *T. lestoquardi* (4.0%) compared to *B. bigemina* (3.0%) and *A. marginale* (3.0%). All nymphs tested negative for any pathogen (Table 5).

### 3.3. Phylogenetic Analysis

The sequences of the ticks in the present study were 100% identical to those of *R. turanicus* reported from Israel (KF958426). We identified three unique sequences of *A. marginale* from three samples, three identical sequences of *B. bigemina* from three samples, and three unique sequences of *T. lestoquardi* from four samples in this study. Phylogenetic analysis revealed *A. marginale* (PP798377) was highly similar to sequences from cattle blood in Thailand, MT796683. Other *A. marginale* sequences (PP798376 and PP798375) were identical to two sequences from cattle ticks in the United States, AF348137, and the Florida strain, AF221693, respectively (Figure 2). *Babesia bigemina* sequences were highly similar to many sequences from cattle blood in Bolivia (LC645222), India (MT322431), Pakistan (KY656456), South Africa (MH257723), and Uganda (KU206297). Lastly, *T. lestoquardi* sequences were highly similar to Iraqi sheep blood, OQ304461, OQ304462, and OQ304458, and Indian sheep blood, MZ665958 (Figure 3).

The sequences obtained in this study were submitted to GenBank under the following accession numbers: PP798375.1, PP798376.1, and PP798377.1 (*A. marginale*), PP809859.1, PP809860.1, and PP809861.1 (*B. bigemina*), and PP798371.1, PP798372.1, PP798373.1 and PP798374.1 (*T. lestoquardi*). All sequences were from tick samples aside from PP798375.1 which was obtained from Indian long-eared hedgehog blood.

## 4. Discussion

Our study only detected the presence of a single tick species, *R. turanicus*, which was found on 25% of collected hedgehogs. *Rhipicephalus* species are potential vectors and reservoirs for several diseases [27,28,29]. Tick prevalence was similar across all three districts: Okara, Pakpattan, and Sahiwal, with no significant differences in the number of ticks collected. Contrarily, research on ectoparasite infestations in European hedgehogs in Iran differed by city, with Urmia City reporting a tick infestation rate of 67.70%, whereas Tabriz City reported an infestation rate of 5.26% [30,31]. This study also identified all ticks as *R. turanicus*. Additionally, a study conducted in Tokat City, Central Anatolia, Turkey, found a higher tick infestation rate of *R. turanicus* at 77.80% [32]. Another study in Pakistan identified *R. turanicus* as the most prevalent tick species, with a 50% infestation rate in free-ranging wild animals [33]. These variations in tick infestation among studies can likely be attributed to differences in geography, temperature, and climate conditions across the studied regions [34]. The two most common tick species observed in European hedgehogs (*Erinaceus europaeus*) in northern Iran were the sheep tick (*Ixodes ricinus*) and the hedgehog tick (*Ixodes hexagonus*), which differ from the ticks, *R. turanicus*, found on the Indian long-eared hedgehog. The geographical spread of *I. ricinus* includes Europe and Northern Africa where *I. hexagonus* includes Europe and central Asia, corresponding with the distribution of their European hedgehog host [34,35]. The exclusion of European hedgehogs in Pakistan may explain the lack of detection of *I. ricinus* and *I. hexagonus* in this study. In the Ninevah district in Iraq, *R. sanguineus*, *Haemaphysalis erinacei*, *Boophilus annulatus*, and *Hyalomma detritum* were found on the long-eared hedgehog [36]. Furthermore, Shubber et al. [37] discovered that *Hyalomma* spp., *R. leparis*, and *R. turanicus* had high infestation rates (76.19%) in long-eared hedgehogs. Additional surveillance on long-eared hedgehogs in Pakistan is needed across a wider geographic range to better understand the diversity of ticks, TBPs, and differences in tick infestation rates. The tick prevalence in livestock in the region of Pakistani Punjab is increasing rapidly, highlighting the need for public health awareness about TBPs and quantifying their distribution in animals [38,39,40].

In this study, female hedgehogs were observed to have a higher tick infestation compared to males; however, no significant association was detected. This finding is consistent with a study conducted in Iraq, where female long-eared hedgehogs (*H. auritus*) exhibited a higher infestation rate (50%) compared to males (44.44%) [41]. Similarly, research conducted by Egyed et al. [42] in Hungary on the northern white-breasted hedgehog (*Erinaceus roumanicus*) in an urban park found that tick prevalence was higher in females (84.21%) than in males (59.65%), whereas no significant differences in sex were found in the hedgehogs of Britain and Germany [43,44]. Variations in sex could likely be explained by behavior variations in male hedgehogs compared to females [45]. In this investigation, young Indian long-eared hedgehogs did not have a statistically significant higher infestation rate compared to adult Indian long-eared hedgehogs. This aligns with no significant difference in tick infestation detected in hedgehogs in Belgium [46]. In contrast, a study on European hedgehogs (*Erinaceus europaeus*) in Germany found subadult hedgehogs had a higher prevalence of tick infestations compared to adults [47]. Variations in tick infestation in hedgehogs of different life stages could be linked to intensified activity levels in younger animals, who may be at a higher risk of pathogen infection due to undeveloped immune systems [48].

From our findings, tick prevalence was similar in Indian long-eared hedgehogs captured from rural areas and urban areas. Hedgehogs naturally inhabit rural areas, where previous research observed European hedgehogs captured from golf courses having higher tick prevalence than those from peri-urban areas [45]. They tend to travel across various agricultural parcels, avoiding areas with significant human populations. The fragmented nature of livestock farms, characterized by multiple walls and fences, forces hedgehogs to cover greater distances with fewer restrictions than in agricultural lands [49,50]. The proximity of agricultural land to natural environments provides a richer variety of habitats compared to non-agricultural areas, often leading to higher species diversity and, consequently, a larger number of ticks [51,52]. We did not detect significant variability in tick infestations between agricultural land and animal farms, which could be explained by our limited sample size.

Detected pathogens belong to the genera of the three most important economic TBDs globally: *Anaplasma*, *Babesia*, and *Theileria.* Among *R. turanicus*, the positive detection rates for *A. marginale* and *Babesia bigemina* were both 2.8%. Both *A. marginale* and *B. bigemina* are endemic in Pakistan [53]. *Anaplasma marginale* has previously been detected from ticks collected from cattle and water buffalo in Pakistan including *R. turanicus*, *R. microplus*, and *H. anatolicum* in Punjab province [53]. In the current study, *A. marginale* was detected in the blood of one hedgehog, with a prevalence rate of 6.3%. Two hedgehogs collected from the Zabol and Bonjar areas of Iran were found to be infected with *Anaplasma* (3.8%), and three *R. turanicus* pools from these infected hedgehogs tested positive for *Anaplasma* spp. [12]. In Punjab province, *A. marginale* and *B. bigemina* have been detected in cattle in a nearby district, Sargodha, Pakistan [54]. The detection of these pathogens in cattle as well as hedgehogs in Pakistan indicates their reservoir potential for tick-borne diseases in this region [55]. This enables ticks to transmit the pathogen from reservoir hosts, such as hedgehogs, to other animals, facilitating the spread of tick-borne pathogen infection. *A. marginale* and *B. bigemina* can be transmitted by multiple tick species, increasing the risk of their spread to livestock. *Theileria lestoquardi* in this study had a prevalence of 3.7% in *R. turanicus* and is the most common *Theileria* spp. found in small ruminants in Africa and Asia [8]. In western Iran, *T. lestoquardi* has been recorded in *R. turanicus*, while in Pakistan and India, it has been detected in goats and sheep [56]. Our study serves as a foundation for future epidemiological studies into ticks, TBPs, and the role hedgehogs play in disease maintenance and potential spillover to livestock and humans.

A limitation of this study is the focus on three specific pathogen taxa, *Anaplasma*, *Babesia*, and *Theileria*. Therefore, we were not able to assess for the presence of other pathogen species that may be transmitted by *R. turanicus* ticks in this region.

## 5. Conclusions

A high prevalence of *R. turanicus* in Indian long-eared hedgehogs was detected in the three study districts in Pakistan. *Rhipicephalus turanicus* harbors numerous pathogens that can cause clinical illness in humans and animals. Due to changes in tick habitat, paired with human, animal, and wildlife behavior changes related to climate change and anthropogenic activities, continued surveillance is necessary to monitor disease dynamics over time. Further epidemiological research is needed in Pakistan outside of Punjab that expands to include other small mammal hosts and tick species of concern, which in time can be used to guide effective prevention and control measures.

## Figures and Tables

**Figure 1 animals-14-03185-f001:**
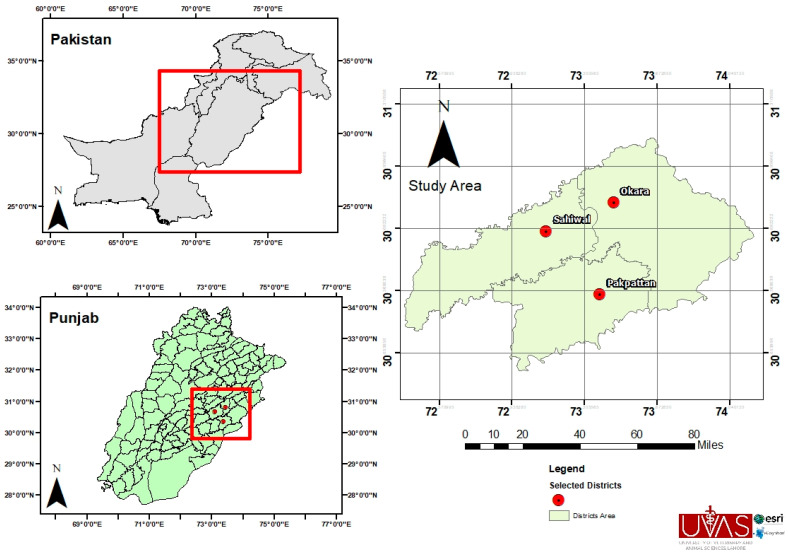
A map showing the area for hedgehog collections in the Sahiwal division, containing three districts: Sahiwal, Okara, and Pakpattan.

**Figure 2 animals-14-03185-f002:**
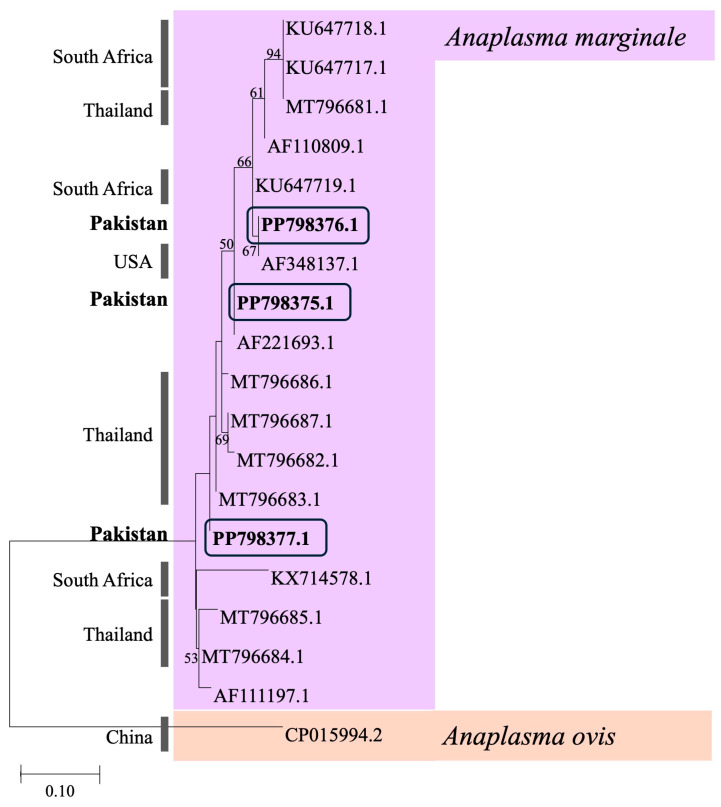
Maximum likelihood tree using the K2 + G model from the msp1b region of *Anaplasma*. Sequences from this study are bolded and outlined by the black rectangles. Corresponding country information of GenBank samples is provided to the left of the tree and pathogen species are color coded.

**Figure 3 animals-14-03185-f003:**
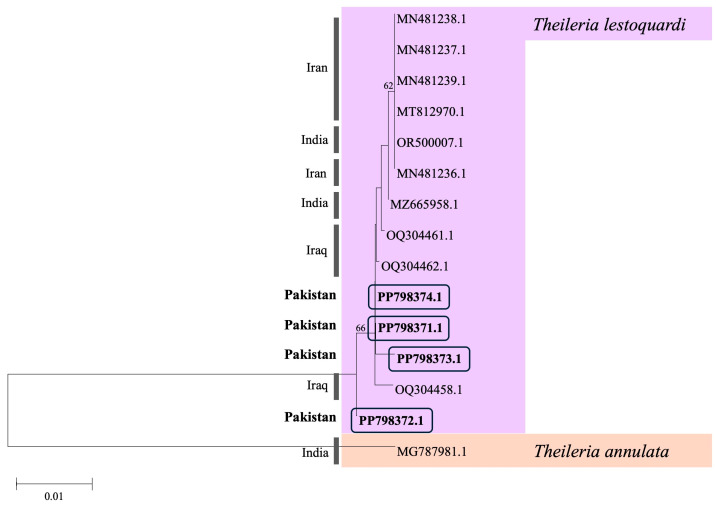
Maximum likelihood tree using the T92 model from the cytb region of *Theileria*. Sequences from this study are bolded and outlined by the black rectangles. Corresponding country information of GenBank samples is provided to the left of the tree and pathogen species are color coded.

**Table 1 animals-14-03185-t001:** List of genus-specific primers used for the detection of TBPs in ticks and blood.

Pathogen	Sequence	Target Region	Size (bp)	Reference
*Anaplasma* spp.	5′-CAGAGCATTGACGCACTACC-3′	msp1b	245	[22]
	5′-TCCAGACCTTCCCTAACTA-3′			
*Babesia* spp.	5′-AGAGGGACTCCTGTGCTTCA-‘3	18s rRNA	321	[23]
	5′-GACGAATCGGAAAAGCCACG-‘3			
*Theileria* spp.	5′-AAGTATAGCAACTGCTTTTGTT-3′	Cyt-b	517	[23]
	5′-TCCTGCCATTGCCAAAAGTC-3′			

**Table 2 animals-14-03185-t002:** Risk factors associated with tick infestation in the Indian long-eared hedgehog population based on Chi-square analysis from Sahiwal division, Punjab province, Pakistan.

Variables	Category	Total Captured	Infested	Prevalence (%)	Chi-Square	*p*-Value
Sex	Female	27	8	29.6	0.465	0.563
	Male	37	8	21.6		
Age	Adult	48	11	22.9	0.505	0.519
	Young	16	5	31.2		
Urbanicity	Rural	38	10	26.3	0.769	0.506
	Urban	26	6	23.1		
Habitat	Agriculture land	35	9	25.7	0.885	0.559
	Animal farm	29	7	24.1		
District	Okara	23	6	26.1	0.030	0.985
	Pakpattan	20	5	25.0		
	Sahiwal	21	5	23.8		

**Table 3 animals-14-03185-t003:** Sex and life stage prevalence of *R. turanicus*.

Variables	Category	Total	Prevalence (%)
Sex	Male	74	67.9
	Female	35	32.1
Life stage	Adult	100	91.7
	Nymph	9	8.3

**Table 4 animals-14-03185-t004:** Infection rates of tick-borne pathogens in hedgehog (*n* = 16) blood using conventional PCR.

Variables	*Anaplasma marginale*
Sex	
Male (*n* = 8)	1 (12.5%)
Female (*n* = 8)	0
Age	
Adult (*n* = 11)	1 (9.1%)
Young (*n* = 5)	0
Urbanicity	
Rural (*n* = 10)	1 (10.0%)
Urban (*n* = 6)	0
Location	
Okara (*n* = 6)	0
Pakpattan (*n* = 5)	0
Sahiwal (*n* = 5)	1 (20.0%)

**Table 5 animals-14-03185-t005:** Infection rate of tick-borne pathogens in ticks (*n* = 109) using conventional PCR.

Variables	Category	*Anaplasma* *marginale*	*Babesia* *bigemina*	*Theileria lestoquardi*	Total
Sex	Male	2 (2.7%)	1 (1.4%)	2 (2.7%)	74
	Female	1 (2.9%)	2 (5.7%)	2 (5.7%)	35
Life stage	Adult	3 (3.0%)	3 (3.0%)	4 (4.0%)	100
	Nymph	0	0	0	9

## Data Availability

The data presented in this study are available within the article.

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
