# Peer review of "Tick Infestation and Molecular Detection of Tick-Borne Pathogens from Indian Long-Eared Hedgehogs (Hemiechinus collaris) in Pakistan"

_animals, 2024, doi:10.3390/ani14223185_

Round 1

Reviewer 1 Report

Comments and Suggestions for Authors

The manuscript concerns ticks feeding on long-eared hedgehog (Hemiechinus collaris) in a region of Pakistan and the identification of three pathogens (Anaplasma marginale, Babesia bigemina i  Theileria lestoquerdi) in the tested ticks. In total. 64 hedgehog individuals were tested, of which 109 tickets were collected. All ticks and hedgehogs’ blood samples were tested for the presence of pathogens belonging to the Anaplasma and Piroplasmida.

The authors do not write whether the primers they used were species- or group-specific. If they were species-specific, screening was limited even further, to only three species.

The authors did not obtain statistical support for the most results on tick prevalence according to various factors (sex, habitat, etc.), which in part may be due to the small number of hedgehogs tested. Despite these limitations, the results obtained are of scientific value, as they involve a host species that is little known in this regard in an area that has not yet been studied. Unfortunately, in this form, the work cannot be published, as it contains many serious errors or inaccuracies. Below are just the most important ones:

The paper is poorly written in English: in many places it is incorrectly scientific (ITS is not a gene, animals have sex and not cultural “gender”, ticks could be vectors or can transmit pathogens (not a carrier) etc. or is not understandable and the editing error is even in the title.

All key words are present in the title or are too general (PCR); instead, Authors can add parasite species detected in the study, the studied area name, and marker genes used to detect pathogens and parasites – it is important for databases indexing papers.

Fig. 2, although very cute, contributes nothing to the results, so it should be removed.

L113 - please add literature references to the keys used here.

Chapter 1.5 needs to be rewritten, because now it is incomprehensible (e.g., “Tick DNA was amplified using the ITS-2 gene” is a mental shortcut”; this DNA was detected by amplification of the intergenic spacer 2 (ITS2) region (it is not a gene, it is a spacer between 5.8S and 28S rRNA genes), values ​​should be separated from units by a space, “master mix” is a kind of lab jargon, etc.

L131 – primers need references.

L140 – which reagents were used in these PCRs? Names and suppliers are needed.

L196 – what do you mean “Positive Rate of TBPs”?

Table 4 is unnecessary, as 0 is indicated in most cells, and this means that its contents can be included in the body text.

The authors should clearly indicate how many types of sequences they found for each marker. In those cases where they found one type and it was 100% identical to a reference sequence in GenBank, there is no need to build phylogenetic trees.

Fig. 3 - the tree is poorly constructed: it should be rooted; on top of that there is an incorrect topology which shows that Rhipicephalus turanicus is not monophyletic; the sequences found in this study should be marked and have GenBank accession numbers added, and there is no information in Material and Methods that NJ method was used for tree construction.

Figs. 4 and 5 in this form are unreadable, they should have stretched branches (this can be easily done in MEGA). Another thing is that they seem unnecessary, because they do not show any geographic structure. Therefore, they can be (after improvement) included in the supplement, while there is no need to present them in body-text.

L281-297 – the first paragraph of the Discussion contributes practically nothing to the discussion, because it provides similar information to the Introduction.

The Discussion is vague and in places unsupported by the results. The authors should refer to the fact that their analysis was focused on specific taxa, leaving out others, such as Borrelia. When discussing the species found on hedgehogs, the authors do not refer to the known ranges of I. ricinus, I. hexagonus and the species they found, and the differences found (not only on this aspect) are quoted with the statement that this is due to climate change and urbanization. Similarly, one cannot agree that Anaplasma and piroplasms are the biggest threat, leaving out Borrelia and rickettsiae.

Comments on the Quality of English Language

The manuscript concerns ticks feeding on long-eared hedgehog (Hemiechinus collaris) in a region of Pakistan and the identification of three pathogens (Anaplasma marginale, Babesia bigemina i  Theileria lestoquerdi) in the tested ticks. In total. 64 hedgehog individuals were tested, of which 109 tickets were collected. All ticks and hedgehogs’ blood samples were tested for the presence of pathogens belonging to the Anaplasma and Piroplasmida.

The authors do not write whether the primers they used were species- or group-specific. If they were species-specific, screening was limited even further, to only three species.

The authors did not obtain statistical support for the most results on tick prevalence according to various factors (sex, habitat, etc.), which in part may be due to the small number of hedgehogs tested. Despite these limitations, the results obtained are of scientific value, as they involve a host species that is little known in this regard in an area that has not yet been studied. Unfortunately, in this form, the work cannot be published, as it contains many serious errors or inaccuracies. Below are just the most important ones:

1.      The paper is poorly written in English: in many places it is incorrectly scientific (ITS is not a gene, animals have sex and not cultural “gender”, ticks could be vectors or can transmit pathogens (not a carrier) etc. or is not understandable and the editing error is even in the title.

2.      All key words are present in the title or are too general (PCR); instead, Authors can add parasite species detected in the study, the studied area name, and marker genes used to detect pathogens and parasites – it is important for databases indexing papers.

3.      Fig. 2, although very cute, contributes nothing to the results, so it should be removed.

4.      L113 - please add literature references to the keys used here.

5.      Chapter 1.5 needs to be rewritten, because now it is incomprehensible (e.g., “Tick DNA was amplified using the ITS-2 gene” is a mental shortcut”; this DNA was detected by amplification of the intergenic spacer 2 (ITS2) region (it is not a gene, it is a spacer between 5.8S and 28S rRNA genes), values ​​should be separated from units by a space, “master mix” is a kind of lab jargon, etc.

6.      L131 – primers need references.

7.      L140 – which reagents were used in these PCRs? Names and suppliers are needed.

8.      L196 – what do you mean “Positive Rate of TBPs”?

9.      Table 4 is unnecessary, as 0 is indicated in most cells, and this means that its contents can be included in the body text.

10.  The authors should clearly indicate how many types of sequences they found for each marker. In those cases where they found one type and it was 100% identical to a reference sequence in GenBank, there is no need to build phylogenetic trees.

11.  Fig. 3 - the tree is poorly constructed: it should be rooted; on top of that there is an incorrect topology which shows that Rhipicephalus turanicus is not monophyletic; the sequences found in this study should be marked and have GenBank accession numbers added, and there is no information in Material and Methods that NJ method was used for tree construction.

12.  Figs. 4 and 5 in this form are unreadable, they should have stretched branches (this can be easily done in MEGA). Another thing is that they seem unnecessary, because they do not show any geographic structure. Therefore, they can be (after improvement) included in the supplement, while there is no need to present them in body-text.

13.  L281-297 – the first paragraph of the Discussion contributes practically nothing to the discussion, because it provides similar information to the Introduction.

14.  The Discussion is vague and in places unsupported by the results. The authors should refer to the fact that their analysis was focused on specific taxa, leaving out others, such as Borrelia. When discussing the species found on hedgehogs, the authors do not refer to the known ranges of I. ricinus, I. hexagonus and the species they found, and the differences found (not only on this aspect) are quoted with the statement that this is due to climate change and urbanization. Similarly, one cannot agree that Anaplasma and piroplasms are the biggest threat, leaving out Borrelia and rickettsiae.

Author Response

Reviewer 1 (red color text)

The manuscript concerns ticks feeding on long-eared hedgehog (Hemiechinus collaris) in a region of Pakistan and the identification of three pathogens (Anaplasma marginale, Babesia bigemina i  Theileria lestoquerdi) in the tested ticks. In total. 64 hedgehog individuals were tested, of which 109 tickets were collected. All ticks and hedgehogs’ blood samples were tested for the presence of pathogens belonging to the Anaplasma and Piroplasmida.

Comment 1: The authors do not write whether the primers they used were species- or group-specific. If they were species-specific, screening was limited even further, to only three species.

Response 1: Thank you for the comment. We used genus-specific primers for the detection of Anaplasma, Babesia, and Theileria and have added this information to the manuscript.

Comment 2: The authors did not obtain statistical support for the most results on tick prevalence according to various factors (sex, habitat, etc.), which in part may be due to the small number of hedgehogs tested. Despite these limitations, the results obtained are of scientific value, as they involve a host species that is little known in this regard in an area that has not yet been studied. Unfortunately, in this form, the work cannot be published, as it contains many serious errors or inaccuracies. Below are just the most important ones:

Response 2: we are grateful for the kind comments of reviewers, we have tried our best to answer the queries of reviewers adequately.

Comment 3: The paper is poorly written in English: in many places it is incorrectly scientific (ITS is not a gene, animals have sex and not cultural “gender”, ticks could be vectors or can transmit pathogens (not a carrier) etc. or is not understandable and the editing error is even in the title.

Response 3: Thank you for your comment. We have replaced the word “gene” with “region”, “gender” with “sex” and “carrier” with “vectors”.

Comment 4: All key words are present in the title or are too general (PCR); instead, Authors can add parasite species detected in the study, the studied area name, and marker genes used to detect pathogens and parasites – it is important for databases indexing papers.

Response 4: Thank you for the suggestion. We have revised the keywords as per the suggestion of the reviewer.

Comment 5: Fig. 2, although very cute, contributes nothing to the results, so it should be removed.

Response 5: Thank you for your comment. Figure 2 has been removed

Comment 6: L113 - please add literature references to the keys used here.

Response 6: Thank you for your comment. The reference is included

Comment 7: Chapter 1.5 needs to be rewritten, because now it is incomprehensible (e.g., “Tick DNA was amplified using the ITS-2 gene” is a mental shortcut”; this DNA was detected by amplification of the intergenic spacer 2 (ITS2) region (it is not a gene, it is a spacer between 5.8S and 28S rRNA genes), values ​​should be separated from units by a space, “master mix” is a kind of lab jargon, etc.

Response 7: Thank you for your comment. The Tick DNA detection section has been revised and all values were separated from units by a space.

Comment 8: L131 – primers need references.

Response 8: Thank you for your comment, the reference has been added.

Comment 9: L140 – which reagents were used in these PCRs? Names and suppliers are needed.

Response 9: Thank you for the comment. We have included the names of the suppliers and locations for all

reagents in parentheses.

Comment 10: L196 – what do you mean “Positive Rate of TBPs”?

Response 10: Thank you for your comment. We replaced “Positive rate of TBPs” with “Prevalence of TBPs”.

Comment 11: Table 4 is unnecessary, as 0 is indicated in most cells, and this means that its contents can be included in the body text.

Response 11: Thank you for the suggestion. However, we have added this table for more clarity of results.

Comment 12: The authors should clearly indicate how many types of sequences they found for each marker. In those cases where they found one type and it was 100% identical to a reference sequence in GenBank, there is no need to build phylogenetic trees.

Response 12: Thank you for your suggestion. We have included the number of unique sequences found for each pathogen in the phylogenetic section (3.3).

Comment 13: Fig. 3 - the tree is poorly constructed: it should be rooted; on top of that there is an incorrect topology which shows that Rhipicephalus turanicus is not monophyletic; the sequences found in this study should be marked and have GenBank accession numbers added, and there is no information in Material and Methods that NJ method was used for tree construction.

Response 13: Thank you for your comment. We believe you are referring to Figure 2. We have decided to remove the figure from the paper as it does not add any additional information to the paper.

Comment 14: Figs. 4 and 5 in this form are unreadable, they should have stretched branches (this can be easily done in MEGA). Another thing is that they seem unnecessary, because they do not show any geographic structure. Therefore, they can be (after improvement) included in the supplement, while there is no need to present them in body-text.

Response 14: Thank you for your comment. We have removed the trees from the manuscript.

Comment 15: L281-297 – the first paragraph of the Discussion contributes practically nothing to the discussion, because it provides similar information to the Introduction.

Reponse 15. Thank you for your comment. We have removed these lines with repetitive information to ensure the discussion is focused on the study findings.

Comment 16: The Discussion is vague and in places unsupported by the results. The authors should refer to the fact that their analysis was focused on specific taxa, leaving out others, such as Borrelia. When discussing the species found on hedgehogs, the authors do not refer to the known ranges of I. ricinus, I. hexagonus and the species they found, and the differences found (not only on this aspect) are quoted with the statement that this is due to climate change and urbanization. Similarly, one cannot agree that Anaplasma and piroplasms are the biggest threat, leaving out Borrelia and rickettsiae.

Response 16. Thank you for your comment. We have revised the discussion to focus on the results of our study. We have included a sentence about the limitation of the taxa we focused on in our study. For the discussion on the hedgehog species we have revised this to include a sentence about the geographic range of I. ricinus and I. hexagonus which are not found in Pakistan.

Reviewer 2 Report

Comments and Suggestions for Authors

This study provides results of a survey in Pakistan on tick-borne pathogens (TBPs) in ticks collected from hedgehogs and in the hedgehogs themselves. Hedgehogs are a potentially important wildlife reservoir of zoonotic TBPs and their role in this regard has only relatively recently been appreciated. Therefore this study represents important surveillance for these pathogens in a lesser studied hedgehog species in an understudied region of the world. This research improves our understanding of the risk that hedgehogs may play in supporting spillover of pathogens to livestock and humans.

I found this to be an interesting and well-written study. Authors detected a number of livestock pathogens in ticks, and in the blood of one hedgehog. The methodology and results are mostly well-described, although some minor improvements can be made to increase clarity. The Discussion makes a number of conclusions based on statistically non-significant results, and this needs to be rewritten to account for these findings - the points in the discussion are still valid but the writing needs to be adjusted to account for the non-significant findings of this study. Limitations should be detailed in Discussion.

Specific comments:

Abstract
1. line 31: would be better to say "...to determine the prevalence and species of TBPs in the blood and ticks..."

2. line 36-38: It is true that these rates were higher in rural vs. urban and agricultural vs. animal farms, but only slightly higher and these were not significantly different, so it is not clear why these two results are highlighted in the abstract. This space may be better used to detail pathogen prevalence in ticks.

3. line 42: since the detected TBPs are primarily of veterinary concern (and this is the journal Animals) I suggest emphasising risk of human and livestock infection.

Introduction
4. line 47: correct "transfer diseases" to "transmit pathogens"

5. line 67: spelling Erinaceidae

Methods
6. line 86: for readers unfamiliar with the region, it may be worth including that Sahiwal division contains three districts Sahiwal, Okara and Pakpattan. Otherwise the map could be slightly confusing.

7. line 108: correct "gender" to "sex". Sex is the preferred term for biological sex, whilst "gender" is more appropriate for psychological and sociocultural traits associated with one's sex (and therefore applies mainly to humans). All mentions of 'gender' in the manuscript should therefore be changed to 'sex'.

8. lines 115-120: I think that it is appropriate to just say that ticks were morphologically identified using taxonomic keys (as stated on lines 113-115) without detailing the characteristics used for identification. These can be found by readers in the cited keys. Therefore I suggest deleting lines 115-120. Also it is not clear why reference no. 20 is cited here as it is not related to tick identification.

9. line 138: state whether these are genus or species-specific primers.

Results
10. lines 178-182: Based on the sample sizes these prevalences are very similar, and accordingly were not statistically different. It would be more appropriate to say "
A similar number of hedgehogs captured in rural areas (26.3%) and urban areas (23.1%) were found infested with ticks. The prevalence of ticks was also similar on hedgehogs found in agricultural land (25.7%) or animal farms (24.1%). Tick prevalence was comparable in all three districts: Okara (26.1%), Pakpattan (25%) and Sahiwal (23.8%)."

11. Table 2: Correct "Infected" to "Infested". Change "gender" to "sex"

12. line 187/Table 3: It is not clear how these statistics were calculated - e.g. it is surprising that 100 is not significantly different from 9. Perhaps extra information on prevalence is required in the table to put this in context?

13. Table 4: It would be useful to repeat in the table how many samples of each were tested, e.g. Male (8) Female (8); Adult (11) Young (5), etc.

14. line 204: state that no nymphs tested positive for any pathogen.

15. line 238: Authors state that tick sequences in the present study were 100% identical to the Israel R. turanicus. However, Figure 3 seems to show significant differences between the 5 UVAS sequences, and 3 are even on a different branch of the tree closer to R. sanguineus from China. Since R. turanicus and R. sanguineus are highly similar tick species, is it possible that some were misidentified, some of the GenBank sequences are from misidentified ticks, or that ITS2 is not a powerful enough sequence to separate these species?

16. line 256: state what UVAS means in the legend.

Discussion
17. line 283: sentence is not clear. Perhaps change to "...
there is a scarcity of information from Pakistan about tick infestation of hedgehogs and the roles of both the vector and host in maintaining TBPs of zoonotic concern."

18. line 285: correct "diseases" to "pathogens"

19. line 299-300: The prevalence was actually very similar across the three districts with no significant difference.

20. line 314, 315: Specify whether H stands for Haemaphysalis or Hyalomma here.

21. line 317-318: Should state that ..."tick prevalence on livestock in the region of Pakistani Punjab is increasing rapidly"

22. line 321: Incorrect - there was not a statistically significant difference between infestation on males and females.

23. line 332: also was not significant.

24. line 334: could this also be related to a less developed immune system in young hedgehogs?

25. line 339: again this was only slightly higher and not significant.

26. line 342: also was comparable in both habitat types and was not a significant result.

27. line 357: yes, but did not vary by a significant amount.

28. line 360: this is a good point, and could be expanded upon further. For example, was sample size a reason why there were no significant differences between the various factors that authors were expecting to cause differences in infestation prevalences by sex, habitat, etc.?

Author Response

Reviewer 2 (Green color text)

General comment: This study provides results of a survey in Pakistan on tick-borne pathogens (TBPs) in ticks collected from hedgehogs and in the hedgehogs themselves. Hedgehogs are a potentially important wildlife reservoir of zoonotic TBPs and their role in this regard has only relatively recently been appreciated. Therefore this study represents important surveillance for these pathogens in a lesser studied hedgehog species in an understudied region of the world. This research improves our understanding of the risk that hedgehogs may play in supporting spillover of pathogens to livestock and humans.

I found this to be an interesting and well-written study. Authors detected a number of livestock pathogens in ticks, and in the blood of one hedgehog. The methodology and results are mostly well-described, although some minor improvements can be made to increase clarity. The Discussion makes a number of conclusions based on statistically non-significant results, and this needs to be rewritten to account for these findings - the points in the discussion are still valid but the writing needs to be adjusted to account for the non-significant findings of this study. Limitations should be detailed in Discussion.

Response: we are thankful to reviewer for valuable comments to improve the write up of the manuscript. Specifically we have revised discussion section.

Specific comments:

Abstract
Comment 1: line 31: would be better to say "...to determine the prevalence and species of TBPs in the blood and ticks..."

Response 1: Thank you for your comment. This has been changed.

Comments 2:  line 36-38: It is true that these rates were higher in rural vs. urban and agricultural vs. animal farms, but only slightly higher and these were not significantly different, so it is not clear why these two results are highlighted in the abstract. This space may be better used to detail pathogen prevalence in ticks.

Response 2: Thank you for the comment. We have removed these sentences to include more pathogen results in the abstract.

Comment 3. line 42: since the detected TBPs are primarily of veterinary concern (and this is the journal Animals) I suggest emphasising risk of human and livestock infection.

Response 3: Thank you for your suggestion. We have adjusted the concluding sentence to reflect this.

Introduction
Comment 4. line 47: correct "transfer diseases" to "transmit pathogens"

Response 4: Thank you for your comment. This has been changed.

Comment 5: line 67: spelling Erinaceidae

Response 5: Thank you for the comment. This has been fixed.

Methods
Comment 6. line 86: for readers unfamiliar with the region, it may be worth including that Sahiwal division contains three districts Sahiwal, Okara and Pakpattan. Otherwise the map could be slightly confusing.

Response 6: Thank you for the suggestion. We have included this in the first line of the methods section.

Comment 7. line 108: correct "gender" to "sex". Sex is the preferred term for biological sex, whilst "gender" is more appropriate for psychological and sociocultural traits associated with one's sex (and therefore applies mainly to humans). All mentions of 'gender' in the manuscript should therefore be changed to 'sex'.

Response 7. Thank you for the comment. We have revised this throughout the manuscript.

Comment 8. lines 115-120: I think that it is appropriate to just say that ticks were morphologically identified using taxonomic keys (as stated on lines 113-115) without detailing the characteristics used for identification. These can be found by readers in the cited keys. Therefore I suggest deleting lines 115-120. Also it is not clear why reference no. 20 is cited here as it is not related to tick identification.

Response 8. Thank you for your comment. We have removed the lines describing the characteristics of morphological identification as we have already cited the taxonomic keys.

Comment 9. line 138: state whether these are genus or species-specific primers.

Response 9. Thank you for the comment. These are genus specific primers and this has been noted in the manuscript.

Results
Comment 10. lines 178-182: Based on the sample sizes these prevalences are very similar, and accordingly were not statistically different. It would be more appropriate to say "A similar number of hedgehogs captured in rural areas (26.3%) and urban areas (23.1%) were found infested with ticks. The prevalence of ticks was also similar on hedgehogs found in agricultural land (25.7%) or animal farms (24.1%). Tick prevalence was comparable in all three districts: Okara (26.1%), Pakpattan (25%) and Sahiwal (23.8%)."

Response 10: Thank you for your suggestion. We have made these changes.

Comment 11. Table 2: Correct "Infected" to "Infested". Change "gender" to "sex"

Response 11: Thank you for the comment. This has been revised.

Comment 12. line 187/Table 3: It is not clear how these statistics were calculated - e.g. it is surprising that 100 is not significantly different from 9. Perhaps extra information on prevalence is required in the table to put this in context?

Response 12: we have simplified the table as suggested by reviewer-1

Comment 13. Table 4: It would be useful to repeat in the table how many samples of each were tested, e.g. Male (8) Female (8); Adult (11) Young (5), etc.

Response 13: Thank you for your comment. We have added a column to the table with the total number tested.

Comment 14. line 204: state that no nymphs tested positive for any pathogen.

Response 14: Thank you for the comment. We have added this line.

Comment 15. line 238: Authors state that tick sequences in the present study were 100% identical to the Israel R. turanicus. However, Figure 3 seems to show significant differences between the 5 UVAS sequences, and 3 are even on a different branch of the tree closer to R. sanguineus from China. Since R. turanicus and R. sanguineus are highly similar tick species, is it possible that some were misidentified, some of the GenBank sequences are from misidentified ticks, or that ITS2 is not a powerful enough sequence to separate these species?

Response 15: We sequenced two tick samples. The sequences of ticks of the present study were 100% identical to those of R. turanicus reported from Israel (GenBank accession no. KF958426). Moreover, phylogenetic tree has been remove on the recommendation of reviewer-3.

Comment 16. line 256: state what UVAS means in the legend.

Response 16: We have removed this Figure from the manuscript so this is no longer needed.

Discussion
Comment 17. line 283: sentence is not clear. Perhaps change to "...there is a scarcity of information from Pakistan about tick infestation of hedgehogs and the roles of both the vector and host in maintaining TBPs of zoonotic concern."

Response 17. Thank you for the comment. We have removed the first paragraph of the discussion as this information is available in the introduction and we can focus more heavily on the results.

Comment 18. line 285: correct "diseases" to "pathogens"

Response 18. Thank you for the comment. We have removed the first paragraph of the discussion as this information is available in the introduction and we can focus more heavily on the results.

Comment 19. line 299-300: The prevalence was actually very similar across the three districts with no significant difference.

Response 19. Thank you for your comment. We have adjusted this sentence.

Comment 20. line 314, 315: Specify whether H stands for Haemaphysalis or Hyalomma here.

Response 20. Thank you for the comment. We have added the respective genera.

Comment 21. line 317-318: Should state that ..."tick prevalence on livestock in the region of Pakistani Punjab is increasing rapidly"

Response 21. Thank you for your suggestion. This has been revised.

Comment 22. line 321: Incorrect - there was not a statistically significant difference between infestation on males and females.

Response 22. Thank you for your comment. We have clarified that we did not find statistically different result in tick infestation in males compared to females.

Comment 23. line 332: also was not significant.

Response 23. Thank you for the comment. We have clarified there was no significant difference.

Comment 24. line 334: could this also be related to a less developed immune system in young hedgehogs?

Response 24. Thank you for your suggestion. We have found evidence that immunity of young hedgehogs could contribute to higher pathogen prevalence.

Comment 25. line 339: again this was only slightly higher and not significant.

Response 25. Thank you for your comment. We have clarified that this was not significant.

Comment 26. line 342: also was comparable in both habitat types and was not a significant result.

Response 26. Thank you for your comment. We have clarified that this was not significant and included this in the discussion of our results.

Comment 27. line 357: yes, but did not vary by a significant amount.

Response 27. Thank you for the comment. We have reworded this.

Comment 28. line 360: this is a good point, and could be expanded upon further. For example, was sample size a reason why there were no significant differences between the various factors that authors were expecting to cause differences in infestation prevalences by sex, habitat, etc.?

Response 28. Although our study provides evidence that ticks from hedgehogs may be a potential vector of tick-borne pathogens, the lack of statistical significance could be attributed to the small sample size.

Reviewer 3 Report

Comments and Suggestions for Authors

The authors have submitted a manuscript on Tick infestation and Molecular Detection of Tick-Borne Pathogens from Indian Long long-eared hedgehog (Hemiechinus collaris) in Pakistan. This is an interesting submission that is likely to benefit many readers interesting the host and vectors. Currently host identification of various pathogens is crucial in predicting outbreaks and epidemics.

General comment

1.  Although the paper sets out with infestation and molecular detection, the paper contains several aspects that include risk factors.

2.  The paper has interesting approach for host tick infestation and infection of vectors. However, the analysis and presentation of the data is not clear to the reader. It is tempting to believe that there was an apparent over analysis of the data. Leaving the data as simplified as possible could make appreciation of the work a lot easier.

3.  The use of terminologies such as infestation percentage (prevalence) or infestation rate may need to be clearly state. The clarity could begin from the methods where a single tick on the Hedgehog would signify infestation. A Host with no tick identified would be considered non infested.

4.  The statistical analysis and the interpretation need clarity and interpreted accordingly.

5.  The section of molecular detection and phylogeny requires some streamlining and interpreted according to the power of the analysis. For instance, what is obtained from the nucleotide comparison or blastn can be contrasted with the information obtained from phylogeny. As it is there is an overlap between these sections.

6.  The discussion can still be reduced and condensed.

Here are a few comments/suggests on the submission.

Specific Comments:

Line 1: There Are two “long” in the title. Is this how it is supposed to be written?

Line 87: it would be helpful for the reader to see the rivers in the map since they are mentioned in the text.

Line 92: would you consider using an alternative term for cultivating e.g. rearing? As the former is generally used in the context of crops and not animals?

Line 94: The citation appears to focus on cattle? Would be helpful to add another for the shoats as well. You may consider replacing the “livestock species” with “cattle breeds”?

Line 95” I suggest including the Sahiwal division in the figure as (A) Pakistan (B) Punjab Province (C) Sahiwal division

Line 104: The description given does not seem to explain much of the Figure 2 e.g. delicate tweezer, damaging ticks mouth? If the space is an issue, I would suggest dropping the figure 2.

Line 106: Was blood collected from animals without infestation if ticks?

Line 108: Kindly clarify how the “ethical consideration” precluded the tissues samples? Was it the aim of the project to collect but you were not allowed or that you plan did not include tissue sample collection. From the way the statement is phrased, it suggests that you had other reasons why tissue samples were not collected:

Line 108: kindly briefly explain the fate of the captured animals? Were they let loose?  

Line 113-119: I suggest clarifying the extent of the use of the Key. Is there need for a key for the following: Sex? Stage of development? Would there be much loss of information if the section was left as a citation or reference to the key? It would be helpful to leave critical aspects that require the reader to appreciate form this section.

Line 122-124: The sentence is quite long plus the citations. Alternative, the “and” (Line 123) can be deleted to begin another sentence.

Line 125: Usually, the citation for the manufacturer would be placed after the kit names instead of the end of the sentence. In which case it suggests that the reference is for the instruction only and not the Kit. Please verify if there is strict journal requirement for this placement.

 Line 126-127: the order of events would be clearer if the measurement is done prior to storage of the purified DNA. Also, the NanoDrop spectrophotometer refers to a particular instrument which must be written according (manufacturer) e.g. ............analysed spectrophotometrically (NanoDrop 2000, Thermo Scientific) or assessed with the Nanodrop 2000 spectrophotometer (Thermo Fisher Scientific, Waltham, MA, USA) 

Line 142: Please clarify the meaning of; “The 5ul PCR product was confirmed with gel electrophoresis” what was being confirmed on the gel? You may consider dropping one of the “gel” in the sentence.

Line 143: Is it fair to say that the visualisation is for both the negative and positive? As mentioned above, the manufacturer is usually placed immediately after the equipment.

Line 143: Note that there may be no need for citation [22] as it doesn’t not contain any of the texted referring to the BioRad gel visualization.

Line 151: Please correct the sentence with citation at the end.

Line 170: the T. Annulata primers are not clearly explained? Hemi-nested?

Line 173: The section on infestation may need clarity. The term infestation and or rate have two different meaning. In this case, I am assuming that the authors used rate synonymous with prevalence. It would be helpful to clearly explain in the methods how the procedure was caried out to arrive at the figures and percentages given.

The general understanding would be that any animal with at least a tick would be considered infested. The rate may now look at how much of ticks were found on each animal. There is risk of losing out on useful information regarding how many ticks were found on an animal. I suggest that the authors revisit the terminologies used here of infestation and or rate.

Line 190-192: Looking at the fluid nature of tick infestation in the environment, you may wonder if there was “over analysis” of the data. This observation is quite evident from the small margins involved in the tables.

Line 193-194: I would suggest that Table 3 be sent to supplements and simply state that the “data was not significant”.

Line 197: I suggest giving the full name of the pathogen at the beginning of the sentence.

Line 218: Table 4 requires some editing and restructuring. Firstly, consider removing the redundant columns. The columns with “0” can be removed or converted to a short paragraph. Consider using the abbreviations for the names of the pathogens in the tables.

Line 198-202: It is not clear whether this description is for Table 4 or 5 since the text does not match with the table contents. Can this be clarified?

Line 200: It is generally agreed that the use of most likely should be restricted to some statistical evaluation rather than percentages. I suggest using other descriptions in place of “more likely”.

Line 237 (Figure 3): How was the 100% identity obtained from the phylogeny analysis? The description given regarding the identity/similarity between R. turanicus

Line 239-244: The description for fig 4 is not clear enough does not clearly speak to the figure. I suggest re-drawing the tree to demonstrate the description being made. As it is, it is difficult to figure out what is being described.

Line 245-251; The description of this tree is similar to Figure 4 and therefore needs more visual clarity to the reader. Alternatively, the closeness of the sequences may not warrant phylogeny analysis or not so many sequences can be used in the analysis. A couple of sequences can be added for demonstration purposes.

Line 284: is the vector’s role referring to the Tick?

Line 286: internationally? Or worldwide?

Line 288: Please provide some examples (citation) for this information.

Line 321: Please compare with Line 187-188. Are they same?

Line 325:I propose using “research conducted by Egyed et al [42] in Hungary....”

Line 321-332: I am of the view that infestation of a host by ticks has more to do with behaviour or other parameters dictating the behaviour in the environment. Could the authors comment on why there is apparent difference in the infestation between males and females? Ending on comparing the various studies may not be sufficient. This also applies to the age variable.

Comments on the Quality of English Language

minor edits

Author Response

Reviewer 3 (brown color text)

The authors have submitted a manuscript on Tick infestation and Molecular Detection of Tick-Borne Pathogens from Indian Long long-eared hedgehog (Hemiechinus collaris) in Pakistan. This is an interesting submission that is likely to benefit many readers interesting the host and vectors. Currently host identification of various pathogens is crucial in predicting outbreaks and epidemics.

 General comment

 Comment 1.  Although the paper sets out with infestation and molecular detection, the paper contains several aspects that include risk factors.

Response 1: we are grateful for kind comment of reviewer

Comment 2.  The paper has interesting approach for host tick infestation and infection of vectors. However, the analysis and presentation of the data is not clear to the reader. It is tempting to believe that there was an apparent over analysis of the data. Leaving the data as simplified as possible could make appreciation of the work a lot easier.

Response 2: we are agree with reviewers comment and make data in table more simple such as Table 3 and 4, now

Comment 3.  The use of terminologies such as infestation percentage (prevalence) or infestation rate may need to be clearly state. The clarity could begin from the methods where a single tick on the Hedgehog would signify infestation. A Host with no tick identified would be considered non infested.

Response 3: we have clarify the criteria of infestation of host in methodology section

Comment 4.  The statistical analysis and the interpretation need clarity and interpreted accordingly.

Response 4. Possible criteria of host positivity and statistical analysis added now

Comment 5.  The section of molecular detection and phylogeny requires some streamlining and interpreted according to the power of the analysis. For instance, what is obtained from the nucleotide comparison or blastn can be contrasted with the information obtained from phylogeny. As it is there is an overlap between these sections.

Response 5: corrected now

Comment 6.  The discussion can still be reduced and condensed.

 Response 6: corrected now

Here are a few comments/suggests on the submission.

Specific Comments:

Comment 7. Line 1: There Are two “long” in the title. Is this how it is supposed to be written?

Response 7. Thank you for your comment. This was a grammatical error that has been corrected.

Comment 8. Line 87: it would be helpful for the reader to see the rivers in the map since they are mentioned in the text.

Response 8. Rivers are not close to this area, that’s why we have deleted this information now

Comment 9. Line 92: would you consider using an alternative term for cultivating e.g. rearing? As the former is generally used in the context of crops and not animals?

Response 9. Thank you for the suggestion. We have reworded this.

Comment 10. Line 94: The citation appears to focus on cattle? Would be helpful to add another for the shoats as well. You may consider replacing the “livestock species” with “cattle breeds”?

Response 10. Citation related to sheep and goat is added now

Comment 11. Line 95” I suggest including the Sahiwal division in the figure as (A) Pakistan (B) Punjab Province (C) Sahiwal division

Response 11: We are grateful for kind suggest of reviewer. However, we have already show in map country, Punjab province and Sahiwal division and its districts

Comment 12. Line 104: The description given does not seem to explain much of the Figure 2 e.g. delicate tweezer, damaging ticks mouth? If the space is an issue, I would suggest dropping the figure 2.

Response 12. Thank you for your suggestion. Figure 2 has been removed.

Comment 13. Line 106: Was blood collected from animals without infestation if ticks?

Comment 13. Blood was collected only animals infested with ticks

Comment 14. Line 108: Kindly clarify how the “ethical consideration” precluded the tissues samples? Was it the aim of the project to collect but you were not allowed or that you plan did not include tissue sample collection. From the way the statement is phrased, it suggests that you had other reasons why tissue samples were not collected:

Response 14. The main purpose of the project was detection of tick-borne pathogens. That why we have deleted text related to tissue samples collection

Comment 15. Line 108: kindly briefly explain the fate of the captured animals? Were they let loose?  

Response 15. Hedgehogs were released back in their natural habitat after collection of samples and data.

Comment 16. Line 113-119: I suggest clarifying the extent of the use of the Key. Is there need for a key for the following: Sex? Stage of development? Would there be much loss of information if the section was left as a citation or reference to the key? It would be helpful to leave critical aspects that require the reader to appreciate form this section.

Response 16. Thank you for the comment. We have removed much of the detailed morphological description and left the citation to the taxonomic key.

Comment 17. Line 122-124: The sentence is quite long plus the citations. Alternative, the “and” (Line 123) can be deleted to begin another sentence.

Response 17. Thank you for your comment. We have split this into two sentences.

Comment 18. Line 125: Usually, the citation for the manufacturer would be placed after the kit names instead of the end of the sentence. In which case it suggests that the reference is for the instruction only and not the Kit. Please verify if there is strict journal requirement for this placement.

Response 18. Thank you for the comment. We have moved manufacturer names directly after the kit name.

 Comment 19. Line 126-127: the order of events would be clearer if the measurement is done prior to storage of the purified DNA. Also, the NanoDrop spectrophotometer refers to a particular instrument which must be written according (manufacturer) e.g. ............analysed spectrophotometrically (NanoDrop 2000, Thermo Scientific) or assessed with the Nanodrop 2000 spectrophotometer (Thermo Fisher Scientific, Waltham, MA, USA)

Response 19. Sentence is rewrite as per suggestion 

Comment 20. Line 142: Please clarify the meaning of; “The 5ul PCR product was confirmed with gel electrophoresis” what was being confirmed on the gel? You may consider dropping one of the “gel” in the sentence.

Response 20. The sentence has been revised for clarity

Comment 21. Line 143: Is it fair to say that the visualisation is for both the negative and positive? As mentioned above, the manufacturer is usually placed immediately after the equipment.

Response 21. Thank you for the comment. We have rephrased this to say visualization of PCR results.

Comment 22. Line 143: Note that there may be no need for citation [22] as it doesn’t not contain any of the texted referring to the BioRad gel visualization.

Response 22. Thank you for your comment. We have removed this reference.

Comment 23. Line 151: Please correct the sentence with citation at the end.

Response 23. Corrected now

Comment 24. Line 170: the T. Annulata primers are not clearly explained? Hemi-nested?

Response 24. Primers sequences has been corrected

Comment 25. Line 173: The section on infestation may need clarity. The term infestation and or rate have two different meaning. In this case, I am assuming that the authors used rate synonymous with prevalence. It would be helpful to clearly explain in the methods how the procedure was caried out to arrive at the figures and percentages given.

The general understanding would be that any animal with at least a tick would be considered infested. The rate may now look at how much of ticks were found on each animal. There is risk of losing out on useful information regarding how many ticks were found on an animal. I suggest that the authors revisit the terminologies used here of infestation and or rate.

Response 25. Thank you for the comment. We have revised this section to remove the word rate as we are referring only to prevalence in our manuscript.

Comment 26. Line 190-192: Looking at the fluid nature of tick infestation in the environment, you may wonder if there was “over analysis” of the data. This observation is quite evident from the small margins involved in the tables.

Response 26. We agree with the reviewer's point and have simplified the data presentation where possible, including revisions to Table 3.

Comment 27. Line 193-194: I would suggest that Table 3 be sent to supplements and simply state that the “data was not significant”.

Response 27. Thank you for the comment. We have simplified Table 3 as suggested by reviewer 1

Comment 28. Line 197: I suggest giving the full name of the pathogen at the beginning of the sentence.

Response 28. Thank you for the suggestion. We have revised this sentence.

Comment 29. Line 218: Table 4 requires some editing and restructuring. Firstly, consider removing the redundant columns. The columns with “0” can be removed or converted to a short paragraph. Consider using the abbreviations for the names of the pathogens in the tables.

Response 29. Thank you for your comment. We have converted Table 4 to a paragraph in the results to

summarize the 1 sample of hedgehog blood positive for A. marginale out of 16 tested samples.

Comment 30. Line 198-202: It is not clear whether this description is for Table 4 or 5 since the text does not match with the table contents. Can this be clarified?

Response 30. The text has been revised, and citations for each table have been included within the manuscript.

Comment 31. Line 200: It is generally agreed that the use of most likely should be restricted to some statistical evaluation rather than percentages. I suggest using other descriptions in place of “more likely”.

Response 31. Sentence has revised now

Comment 32. Line 237 (Figure 3): How was the 100% identity obtained from the phylogeny analysis? The description given regarding the identity/similarity between R. turanicus

Response 32. Thank you for the comment. We used the ITS2 region to identify the tick species moleculary and have removed Figure 2 with the R. turanicus phylogenetic tree as it did not add information to the paper.

Comment 33. Line 239-244: The description for fig 4 is not clear enough does not clearly speak to the figure. I suggest re-drawing the tree to demonstrate the description being made. As it is, it is difficult to figure out what is being described.

Response 33. Thank you for your comment. We have removed figure 4 from the manuscript.

Comment 34. Line 245-251; The description of this tree is similar to Figure 4 and therefore needs more visual clarity to the reader. Alternatively, the closeness of the sequences may not warrant phylogeny analysis or not so many sequences can be used in the analysis. A couple of sequences can be added for demonstration purposes.

Response 34. Thank you for your comment. We have removed figure 5 from the manuscript.

Comment 35. Line 284: is the vector’s role referring to the Tick?

Response 35. Thank you for your comment. We have removed the first paragraph of the discussion with

redundant information from the introduction.

Comment 36. Line 286: internationally? Or worldwide?

Response 36. Thank you for the comment. See note about line 284.

Comment 37. Line 288: Please provide some examples (citation) for this information.

Response 37. Thank you for the comment. See note about line 284.

Comment 38. Line 321: Please compare with Line 187-188. Are they same?

Response 38. These are not same

Comment 39. Line 325:I propose using “research conducted by Egyed et al [42] in Hungary....”

Response 39. Thank you for the comment. This has been edited.

Comment 40. Line 321-332: I am of the view that infestation of a host by ticks has more to do with behaviour or other parameters dictating the behaviour in the environment. Could the authors comment on why there is apparent difference in the infestation between males and females? Ending on comparing the various studies may not be sufficient. This also applies to the age variable.

Response 40. Thank you for comment. We have reworded this section of the discussion to focus on tick

behavior and other potential hypotheses accounting for these differences.

Round 2

Reviewer 1 Report

Comments and Suggestions for Authors

Chapter 3.3 Phylogenetic analysis is incomprehensible and does not refer to any results (= phylogenetic tree) and therefore makes no sense. All this chapter is badly written. For example, sequences of "a greater genetic diversity" means that they are variable, however, the authors found only one variant of the B. bigemina sequence. But this is only one example, the whole chapter should be re-edited.

Comments on the Quality of English Language

The paper is still written in unintelligible English; this is especially true for the results of the sequence analysis.

Author Response

Reviewer 1 (blue color text)

Comment 1. Chapter 3.3 Phylogenetic analysis is incomprehensible and does not refer to any results (= phylogenetic tree) and therefore makes no sense. All this chapter is badly written. For example, sequences of "a greater genetic diversity" means that they are variable, however, the authors found only one variant of the B. bigemina sequence. But this is only one example, the whole chapter should be re-edited.

Response 1. Thank you for your comment. We have added in two trees for Anaplasma and Theileria to expand upon the phylogenetic section. The Babesia sequences were identical to each other and highly similar to many GenBank sequences so we did not include them in a tree as it would not add additional information. This section (3.3) has also been revised.

Comments on the Quality of English Language

Comment 3. The paper is still written in unintelligible English; this is especially true for the results of the sequence analysis.

Response 3. Thank you for the comment. We have reviewed the paper for grammatical errors and fixed any mistakes.

Reviewer 2 Report

Comments and Suggestions for Authors

The authors have really improved this paper with their revision, especially the Discussion. I recommend that the paper be accepted following correction of the following minor errors:

1. line 32: correct "longed-eared" to "long-eared"

2. line 44: correct "proving" to "providing"

3. line 86: correct to "(which contains three districts Sahiwal, Okara and Pakpattan)

4. line 144: Microgen, Korea doesn't need to be in brackets.

5. line 184: correct grammar errors in Table legend "Risk factors associated with tick infestation in the Indian long-eared hedgehog population..."

6. line 189: subtitle should be numbered 3.2

7. line 266: include the scientific name of the northern white-breasted hedgehog.

8. line 274: correct grammar, change "Contrastly" to "In contrast,"

9. line 327: correct "in" to "on" in the sentence - "A high prevalence of R. turanicus on Indian long-eared hedgehogs.."

Comments on the Quality of English Language

A few grammatical errors are present in the text, which I have noted in "Comments and Suggestions for Authors
"

Author Response

Reviewer 2 (red color text)

The authors have really improved this paper with their revision, especially the Discussion. I recommend that the paper be accepted following correction of the following minor errors:

Comment 1. line 32: correct "longed-eared" to "long-eared"

Response 1. Thank you for your comment. We have revised this.

Comment 2. line 44: correct "proving" to "providing"

Response 2. Thank you for the comment. This has been fixed.

Comment 3. line 86: correct to "(which contains three districts Sahiwal, Okara and Pakpattan)

Response 3. Thank you for your comment. We have changed the order of this sentence.

Comment 4. line 144: Microgen, Korea doesn't need to be in brackets.

Response 4. Thank you for the comment. We have fixed this.

Comment 5. line 184: correct grammar errors in Table legend "Risk factors associated with tick infestation in the Indian long-eared hedgehog population..."

Response 5. Thank you for the comment. This has been adjusted throughout the paper.

Comment 6. line 189: subtitle should be numbered 3.2

Response 6. Thank you for the comment. This has been changed.

Comment 7. line 266: include the scientific name of the northern white-breasted hedgehog.

Response 7. Thank you for the comment. This has been changed.

Comment 8. line 274: correct grammar, change "Contrastly" to "In contrast,"

Response 8. Thank you your comment. This has been fixed.

Comment 9. line 327: correct "in" to "on" in the sentence - "A high prevalence of R. turanicus on Indian long-eared hedgehogs.."

Response 9. Thank you your comment. We have changed this.

Comments on the Quality of English Language

Comment 10. A few grammatical errors are present in the text, which I have noted in "Comments and Suggestions for Authors
"

Response 10. Thank you for the comment. We have reviewed the manuscript for additional grammar and made necessary changes.

Reviewer 3 Report

Comments and Suggestions for Authors

Dear Authors:

I wish to recognise the extensive revision to the manuscript. The change is noticeable.

Some minor comments:

Line 39:  was Vs were

Line 40: "....tick's samples"? possessive?

 Line 42: "......livestock animal’s"? as above?

Line 43: ".....spillover "of" such....)

Line 190: Suggestion: "Anaplasma marginale was detected an adult male hedgehog captured from a rural area of district Sahiwal..."

Line 197: "No nymphs...."

Comments on the Quality of English Language

There maybe some grammatical errors/typos that could have arisen from the edits of the manuscript. Some edits for clarity may be advised. 

Author Response

Reviewer 3

Dear Authors:

I wish to recognise the extensive revision to the manuscript. The change is noticeable.

Some minor comments:

Comment 1. Line 39:  was Vs were

Response 1. Corrected now

Comment 2. Line 40: "....tick's samples"? possessive?

Response 2. Thank you for your comment. This has been adjusted.

 Comment 3. Line 42: "......livestock animal’s"? as above?

Response 3. Thank you for your comment. This apostrophe has been removed.

Comment 4. Line 43: ".....spillover "of" such....)

Response 4. Thank you for the comment. We have fixed this line.

Comment 5. Line 190: Suggestion: "Anaplasma marginale was detected an adult male hedgehog captured from a rural area of district Sahiwal..."

Response 5. Thank you for your comment. We have edited this line.

Comment 6. Line 197: "No nymphs...."

Response 6. Thank you for your comment. We have edited this line.

Comments on the Quality of English Language

Comment 7. There maybe some grammatical errors/typos that could have arisen from the edits of the manuscript. Some edits for clarity may be advised. 

Response 7. Thank you for the comment. We have read through the manuscript for any additional grammatical edits.

Round 3

Reviewer 1 Report

Comments and Suggestions for Authors

The MS has been improved sufficiently.